# Improvement of PR8-Derived Recombinant Clade 2.3.4.4c H5N6 Vaccine Strains by Optimization of Internal Genes and H103Y Mutation of Hemagglutinin

**DOI:** 10.3390/vaccines8040781

**Published:** 2020-12-20

**Authors:** Se-Hee An, Seung-Min Hong, Seung-Eun Son, Jin-Ha Song, Chung-Young Lee, Jun-Gu Choi, Youn-Jeong Lee, Jei-Hyun Jeong, Jun-Beom Kim, Chang-Seon Song, Jae-Hong Kim, Kang-Seuk Choi, Hyuk-Joon Kwon

**Affiliations:** 1Laboratory of Avian Diseases, College of veterinary medicine, Seoul National University, Seoul 08826, Korea; eepdl1201@snu.ac.kr (S.-H.A.); topkin@snu.ac.kr (S.-M.H.); arbre04@snu.ac.kr (S.-E.S.); sjh1243@snu.ac.kr (J.-H.S.); Chung-Young.Lee@emory.edu (C.-Y.L.); kchoi0608@snu.ac.kr (K.-S.C.); 2Research Institute for Veterinary Science, College of Veterinary Medicine, Seoul National University, Seoul 08826, Korea; kimhong@snu.ac.kr; 3Avian Disease Division, Animal and Plant Quarantine Agency, 177, Hyeoksin 8-ro, Gyeongsangbuk-do 39660, Korea; happythomas@korea.kr (J.-G.C.); leeyj700@korea.kr (Y.-J.L.); 4Laboratory of Avian Diseases, College of Veterinary Medicine, Konkuk University, Seoul 05029, Korea; nar21ss@hanmail.net (J.-H.J.); kjb900809@naver.com (J.-B.K.); songcs@konkuk.ac.kr (C.-S.S.); 5Farm Animal Clinical Training and Research Center (FACTRC), Institutes of Green-Bio Science & Technology, Seoul National University, Kangwon-do 25354, Korea

**Keywords:** clade 2.3.4.4c H5N6 vaccine, antigen productivity, T cell epitopes, heat/acid stability

## Abstract

Clade 2.3.4.4c H5N6 avian influenza A viruses (AIVs) may have originally adapted to infect chickens and have caused highly pathogenic avian influenza (HPAI) in poultry and human fatalities. Although A/Puerto Rico/8/1934 (H1N1) (PR8)-derived recombinant clade 2.3.4.4c H5N6 vaccine strains have been effective in embryonated chicken eggs-based vaccine production system, they need to be improved in terms of immunogenicity and potential mammalian pathogenicity. We replaced the PB2 gene alone or the PB2 (polymerase basic protein 2), NP (nucleoprotein), M (matrix protein) and NS (non-structural protein) genes together in the PR8 strain with corresponding genes from AIVs with low pathogenicity to remove mammalian pathogenicity and to match CD8+ T cell epitopes with contemporary HPAI viruses, respectively, without loss of viral fitness. Additionally, we tested the effect of the H103Y mutation of hemagglutinin (HA) on antigen productivity, mammalian pathogenicity and heat/acid stability. The replacement of PB2 genes and the H103Y mutation reduced the mammalian pathogenicity but increased the antigen productivity of the recombinant vaccine strains. The H103Y mutation increased heat stability but unexpectedly decreased acid stability, probably resulting in increased activation pH for HA. Interestingly, vaccination with inactivated recombinant virus with replaced NP, M and NS genes halted challenge virus shedding earlier than the recombinant vaccine without internal genes replacement. In conclusion, we successfully generated recombinant clade 2.3.4.4c H5N6 vaccine strains that were less pathogenic to mammals and more productive and heat stable than conventional PR8-derived recombinant strains by optimization of internal genes and the H103Y mutation of HA.

## 1. Introduction

Highly pathogenic avian influenza A viruses (HPAIVs) are highly contagious pathogens causing serious economic and public health problems [1,2]. The clade 2.3.4.4 HPAIVs have diversified into multiple distinct subgroups (a–h), and clade 2.3.4.4c H5N6 HPAIVs caused major endemics in China during years 2014–2016 and outbreaks in Korea in 2016 [3,4,5,6]. Additionally, the clade 2.3.4.4c H5N6 HPAIVs caused infections of mammals including humans, and most showed a single amino acid (133, H3 numbering) deletion in the receptor-binding site of hemagglutinin (HA) and an 11 amino acid deletion in the stalk of neuraminidase (NA), reflecting a balance of HA and NA activity and adaptation to chickens [7].

The high-growth A/Puerto Rico/8/1934 (H1N1) (PR8) strain is high-yielding in embryonated chicken eggs (ECEs), and PR8-based 6+2 recombinant vaccine strains have been used in countries where highly pathogenic avian influenza (HPAI) became endemic [8,9,10]. However, the internal genes of PR8 have already acquired multiple mutations related to mammalian pathogenicity, and conventional PR8-derived recombinant vaccine strains against HPAI can potentially acquire mammalian pathogenicity in vitro during reverse genetics procedures [11,12]. Therefore, studies to find internal genes that are less pathogenic to mammals while maintaining viral replication efficiency have been continued by replacing internal genes of PR8 with single PB2 or NS genes and all internal genes of low pathogenic avian influenza viruses (LPAIVs) [12,13,14].

HPAI vaccines have been developed to induce humoral immunity against HA and NA, and oil emulsion of formaldehyde-inactivated whole virus has been a commonly chosen formula for animal vaccines. Major CD8+ T cell epitopes for cellular immunity have been embedded mainly in nucleocapsid (NP) and matrix (M) proteins and are more conserved than B cell epitopes of HA and NA used to induce heterosubtypic protection [15,16]. Recently, cellular immunity has been demonstrated in chickens that were inoculated with beta-propiolactone (BPL)-inactivated oil emulsion vaccine. BPL-inactivation did not hinder the endocytosis of viruses or invasion of viral RNPs (ribonucleoproteins) into cytoplasm to expose CD8+ T cell epitopes for presentation by MHC (major histocompatibility complex) class I molecules [17]. Therefore, matching CD8+ T cell epitopes of vaccine strains with those of contemporary field viruses may improve heterosubtypic cross protection efficacy of vaccine strains. The anchor residues interacting with the chicken MHC class I molecules are already known, and CD8+ T cell epitopes in internal proteins can be predicted through these residues [18,19]

ECEs have long been used as seasonal flu and AI vaccine production platforms due to their cost-effectiveness. The amount of viral antigen is usually proportional to virus titers, and vaccine strains showing high replication efficiency in embryonated chicken eggs are preferable. The H103Y mutation has been reported not only to increase mammalian pathogenicity of H5N1 HPAIVs, thermostability, and fusion of HA at low pH environment but also viral replication efficiency in ECEs [20,21,22]. Recently, we adopted H103Y in the clade 2.3.4.4a H5N8 vaccine strain to improve its immunogenicity by increasing viral replication efficiency and thermostability [13]. Although we successfully generated a mammalian nonpathogenic recombinant clade 2.3.4.4a H5N8 vaccine strain by replacing PR8 PB2 with the nonpathogenic PB2 gene, the H103Y mutation in itself increased the pathogenicity to replicate more efficiently in Madin-Darby canine kidney (MDCK) and A549 cells [13].

In this study, we aimed to improve conventional PR8-derived recombinant 2.3.4.4c H5N6 vaccine strains in terms of antigen productivity, reduced mammalian pathogenicity, protection efficacy and antigenic stability. To these ends we generated recombinant vaccine strains by replacing the PB2, NP, M and NS genes of PR8 with corresponding genes of LPAIVs and the H103Y mutation of HA and experimentally demonstrated reduced mammalian pathogenicity and improved antigen productivity, protection efficacy and thermostability of the vaccine strains in vitro and in vivo.

## 2. Materials and Methods

### 2.1. Viruses, Plasmids, Eggs and Cells

All genome segments were cloned into pHW2000 reverse genetics vector for recombinant virus generation [23]. The 293T cells were purchased from the Korean Collection for Type Cultures (KCTC, Daejeon, Korea) were maintained in DMEM (Dulbecco’s Modified Eagle’s Medium) supplemented with 10% fetal bovine serum (FBS; Life Technologies, Carlsbad, CA, USA) and used for transfection. Viruses used in this study were propagated in specific-pathogen-free (SPF) ECEs (VALO BioMedia GmbH, Osterholz-Scharmbeck, Lower Saxony, Germany). MDCK cells and A549 cells were purchased from KCTC and maintained in DMEM and DMEM/F12, respectively, supplemented with 10% FBS (Life technologies, Carlsbad, CA, USA). 

HA and NA genomes of recombinant H5N6 virus were synthesized in the sequence most frequent among clade 2.3.4.4c H5N6 isolates during the years 2014–2016 (*n* = 73) (Cosmo Genetech, Seoul, Korea), with a change in cleavage site from a polybasic amino acid (RERRKR) site to the monobasic amino acid (ASGR) site of the Korean H9N2 LPAI virus, A/chicken/Korea/KBNP-*0028*/00(*H9N2*) (*0028*) [24]. Internal genes of PR8, LPAIV H9N2 01310, *0028* and LPAI A/wild duck/Korea/SNU50-5/2009(H5N1) (SNU50-5) were used to generate recombinant H5N6 strains possessing internal proteins of PR8 or avian IAVs (Influenza A viruses) [12,22,23]. Recombinant H5N8 viruses (rH5N8 and rH5N8-H103Y) used in pH stability tests in ECEs were the same viruses employed in our previous study on the clade 2.3.4.4a H5N8 vaccine [13]. The rH5N1-310PB2 and rH5N1-H103Y-310PB2 viruses were generated with the HA and NA genomes of clade 2.3.2.1c H5N1 HPAIV, A/Mandarin duck/Korea/K10-483/2010 (K10-483) based on PB2 of 01310 and five internal genes of PR8 [23]. The heat-stable H103Y mutation in the HA gene was generated using a Muta-Direct™ site-directed mutagenesis kit (iNtRON Biotechnology, Seongnam, Gyeonggi, Korea) and a mutagenesis primers set (Table 1).

To compare sequence identities between the internal genes used in this study and the genomes of HPAIVs, genomic sequences of wild-type HPAI H5N1, H5N8 and H5N6 viruses isolated during the years 2018–2020 were retrieved from the GISAID (Global Initiative for Sharing All Influenza Data) Epiflu database, and the BioEdit program (v7.2.5) was used. CD8+ T cell epitopes were predicted as previously described [18,19]. 

### 2.2. Generation of Recombinant H5N6 Strain 

The bidirectional pHW2000 vector and PR8-based reverse genetics system were used to generate a recombinant H5N6 virus with few modification, as described previously [13,23]. Confluent 293T cells (KCTC) in a 6-well plate were transfected with 300 ng of each plasmid comprising eight genomic segments mixed in Opti-MEM (Life Technologies) using Plus-reagents and Lipofectamine 2000 (Life Technologies), according to the supplier’s instructions. After 16 h, 1 mL of Opti-MEM and 4 µg/well of L-1-tosylamido-2-phenylethyl chloromethyl ketone (TPCK)-treated trypsin (Sigma-Aldrich, St. Louis, MO, USA) were added, and the plate was incubated for 24 h. Two hundred microliters of each supernatant were inoculated into 10-day-old SPF ECE, and recombinant virus in harvested allantoic fluid was identified after 72 h by a hemagglutination assay using 1% (*v*/*v*) chicken red blood cells (RBCs), and the full genome sequences were confirmed by PCR. Briefly, RNA was extracted from harvested allantoic fluid using Viral Gene-spin™ Viral DNA/RNA Extraction Kit (iNtRON Biotechnology, Seongnam, Gyeonggi, Korea) and cDNA was synthesized by TOPscript™ cDNA Synthesis kit (Enzynomics, Daejeon, Korea). Full genomes were amplified using universal primer sets previously described [23]. 

### 2.3. Virus Titration in ECEs

Confirmed recombinant virus (CE1) was propagated in 10-day-old SPF ECEs at 37 °C for 72 h, the harvested CE2 virus was titrated as the 50% chicken embryo infectious dose (EID_50_), and aliquots of allantoic fluid were stored at −80 °C until use. Serial dilutions of CE2 from 10^−1^ to 10^−9^ were inoculated into five 10-day-old SPF ECEs, and the presence of virus in each dilution was confirmed by hemagglutination assay after a 72-h incubation. EID_50_ of each virus was calculated by the Spearman–Karber method [25]. For comparisons of the replication efficiency in the ECEs, 100 EID_50_/0.1 mL dilution of each CE2 virus was injected into five 10-day-old SPF ECEs, and the titer of replicated viruses in allantoic fluid was measured as EID_50_.

### 2.4. Growth Kinetics of the Recombinant H5N6 Strain in Mammalian Cell Lines

MDCK and A549 monolayers in a 12-well plate were infected with 5 × 10^5^ EID_50_/0.5 mL of virus diluted in DMEM with 1 µg/mL TPCK-trypsin. The viral solution was removed after a 1-h incubation, and fresh medium containing 1 µg/mL TPCK-trypsin was added. During incubation in a CO_2_ incubator at 37 °C for 72 h, supernatant was obtained every 24 h. The 50% tissue culture infective dose (TCID_50_) of virus at each time point was measured by inoculation of a 10-fold diluent of cell supernatant into MDCK cells in a 96-well plate and calculated by the Spearman–Karber method. 

### 2.5. Vaccine Efficacy Testing in Chickens and Ducks

For inactivated vaccine preparation, 100 EID_50_ of each recombinant H5N6 virus was inoculated into five 10-day-old SPF ECEs, and allantoic fluid was harvested after a 72-h incubation at 37 °C. Harvested fluid was inactivated by mixing it with 0.1 M binary ethylenimine (BEI) prepared by mixing 0.041 g of 2-bromoethylamine hydrobromide (Sigma-Aldrich) with 2 mL of 0.175 N sodium hydroxide (Sigma-Aldrich) and at 37 °C for 24 h, and BEI was neutralized by 1 M sodium thiosulfate at the end of each reaction. Undiluted BEI-treated viral preparations were inoculated into 10-day-old SPF ECEs to confirm inactivation of the viruses. Three volumes (1.5 mL) of each inactivated virus (or allantoic fluid for the mock group) were mixed with 7 volumes (3.5 mL) of ISA 70 (SEPPIC, Courbevoie, France) to produce an oil-emulsion vaccine. Three-week-old SPF chickens (Namduck Sanitek, Icheon, Korea) (*n* = 9 in the first experiment and *n* = 8 in the second experiment) and 2-week-old ducks (*n* = 10) were vaccinated subcutaneously with 0.5 mL of each of oil-emulsion inactivated vaccine. Serum samples from each group were collected 0, 3 and 4 weeks postvaccination (wpv) to measure the serum antibody titers. An intranasal challenge experiment on chickens with 10^6^ EID_50_/0.1 mL of A/Mandarin duck/Korea/K16-187-3/2016 (H5N6) was approved and conducted in a biosafety level 3 facility at Konkuk University (IACUC-KU18179). Oropharyngeal and cloacal swab samples were collected 1, 3, 5 and 7 days after the challenge, and viral shedding in oropharynx and cloaca was evaluated by RT-PCR using an Applied Biosystems 7500 Real-Time PCR System (Life Technologies) as previously described [26]. 

### 2.6. Hemagglutination Inhibition (HI) Test

Serum samples of vaccinated chicken groups were treated at 56°C for 30 min, and serum samples from the duck groups were mixed with 3 volumes of each receptor-destroying enzyme II (RDE II; Denka Seiken Co., Ltd., Tokyo, Japan) for 24 h before heat treatment to inhibit innate duck serum inhibitors. HI assays were performed based on the WHO Manual on Animal Influenza Diagnosis and Surveillance with modification. Briefly, 25 μL of each serum sample was 2-fold diluted serially with phosphate-buffered saline (PBS), and the same volume of 4 hemagglutination titers [27] of homologous recombinant H5N6 virus was added. After a 40-min incubation at 4 °C, 25 μL of 1% chicken RBCs were added to initiate the hemagglutination reaction, and the HI titer was recorded after 40 min at 4 °C. 

### 2.7. Quantification of Purified Total Viral Proteins and SDS-PAGE 

For the quantitative measurement of viral particles in allantoic fluid, recombinant virus was purified using OptiPrep™ density gradient medium (Sigma-Aldrich) and an Optima™ XL-100K Ultracentrifuge (Beckman Coulter, Brea, CA, USA). Inoculations of 100 EID_50_/0.1 mL of each recombinant virus was performed with eight 10-day-old SPF ECEs, and allantoic fluid was harvested after a 3-day incubation at 37 °C. HAT (hemagglutinin titer) of the harvested fluid was titrated as described above, and the harvested fluids were pooled for total protein purification. Fifteen milliliters of pooled allantoic fluid was clarified at 13,000 rpm for 10 min, and the supernatant containing virus was layered on a 30% density medium in a thick-walled tube used for an SW 70 Ti rotor (Beckman) and centrifuged at 35,000 rpm for 2 h. The sedimented virus above the 30% density medium was transferred to another thick-walled tube with 9 mL of PBS and centrifuged at 30,000 rpm for 1 h. The supernatant was discarded, and the pelleted virion was resuspended in 100 μL of PBS. The quantity of the purified viral protein was measured with a SMART™ BCA assay kit (iNtRON Biotechnology, Seongnam, Gyeonggi, Korea), following the manufacturer’s instructions. Purified viral protein was treated with PNGase F (N-glycosidase F; New England Biolabs, Ipswich, MA, USA) to deglycosylate viral protein, following the manufacturer’s instructions. To confirm the purified viral protein, 10 μL of purified total viral protein and deglycosylated viral protein were denatured with Protein 5X sample buffer (ELPIS BIOTECH, Daejeon, Korea) for 5 min at 95 °C, and SDS-PAGE was performed with NuPAGE 4%–12% Bis-Tris Protein Gels (Life Technologies). The separated proteins were stained with Coomassie Brilliant Blue G-250 staining solution (Biosesang, Seongnam, Gyeonggi, Korea) for 1 h and destained with a solution consisting of 10% ethanol and 7.5% acetic acid. 

### 2.8. Heat and Low pH Stability Testing

To compare inactivation of HA protein of recombinant viruses after heat treatment, a heat stability test was conducted, as previously described with modifications [13]. Sixteen (2^4^) HAT diluents of each virus were aliquoted into 1.5-mL tubes for heat treatment at 55 °C for different times (0, 15, 30, 45, 60, 90 min, 2, 3 and 6 h), and HAT of each heat-treated virus was measured again. 

Virus inactivation tests were performed following the method of Baumann et al., 2016 [28]. Briefly, pH buffer (pH 4.6–6.2) was prepared using PBS and 0.1 M citric acid (Sigma-Aldrich) and 10^8^ EID_50_ of each virus was diluted 1/100 with pH buffer and incubated at 37 °C for 2 h. The pH treated virus was diluted 1/100 with infection medium (PBS for ECE inoculation) and inoculated into MDCK monolayers in a 12-well plate and three 10-day-old SPF ECEs. After a 1-h incubation in MDCK cells, the infected virus was replaced with fresh infection medium, and the viral titers of cell supernatant and allantoic fluid at 72 h post inoculation (hpi) were measured as TCID_50_ and HA. 

### 2.9. Statistical Analysis

Statistical significance of differences between experimental groups were evaluated by one-way analysis-of-variance (log-rank test, 95% confidence intervals) (IBM SPSS Statistics, Armonk, NY, USA), and *p* < 0.05 was defined as statistically significant. 

## 3. Results

### 3.1. Comparison of the Predicted CD8+ T Cell Epitopes in the NP and M Proteins 

The identified CD8+ T cell epitopes in the NP and M1 proteins of the challenge virus were seven and four, respectively, and the amino acid sequences were compared with corresponding sequences of internal gene donor viruses (50–5, 01310, 0028 and PR8) and H5Nx (H5N1, H5N8 and H5N6) viruses isolated during the years 2018–2020 (Table 2). The four CD8+ T cell epitopes of SNU50-5 and 0028 were perfectly matched with those of the challenge virus except for one (76–83 and 368–375, respectively) with a single amino acid mismatch. Moreover, 01310 showed a single amino acid mismatch in two epitopes (347–354 and 368–375), but PR8 exhibited single or multiple mismatches including anchor amino acids in all epitopes. One epitope of challenge virus M1 was identical to those of SNU50-5, 01310 and 0028, but none were identical to those of PR8. The NP and M1 CD8+ T cell epitopes of the H5Nx viruses differed from each other by 0 to 2 amino acids. The complete amino acid sequence identities of the NP and M1 in the strains that were compared are shown in Appendix A. The amino acid sequence identities of SNU50-5 NP and M1 to H5Nx were higher (by 98.3%–99.4% and 94.0%–96.1%, respectively) than those for PR8 (92.8%–94.0% and 91.7%–92.8%, respectively), 01310 (97.4%–98.0% and 93.9%–95.9%, respectively) and 0028 (98.1%–98.6% and 93.9%–94.9%, respectively). The amino acid sequence of the challenge virus M2e was also compared, and each virus showed differences of 0 to 5 amino acids (Appendix A).

### 3.2. Generation and Replication Efficiency of CD8+ T Cell Epitope-Matched Recombinant H5N6 Strain

We selected the SNU50-5 NP gene for better CD8+ T cell epitope matching, but we selected the 01310 M gene without further considering the T or B cell epitope matching due to viral fitness. Although the effects of the 01310 PB2 and 0028 NS genes on decreased mammalian pathogenicity and increased viral yield in the ECEs have already been verified in our previous studies, the combination of 01310 PB2, SNU50-5 NP, 01310 M and 0028 NS internal genes was first tested in this study. PR8-derived recombinant clade 2.3.4.4c H5N6 strains possessing all the internal genes of PR8 (rH5N6), with internal genes replaced by 01310 PB2 (rH5N6-310PB2) and with internal genes replaced by 01310 PB2, SNU50-5 NP, 01310 M and 0028 NS (rH5N6-IG) were generated (Table 3). The virus titers of rH5N6, rH5N6-310PB2 and rH5N6-IG were 9.08 ± 0.14, 9.33 ± 0.29 and 9.25 ± 0.25 EID_50_/_mL_, and there were no significant differences among them (*p* > 0.05, Table 3). rH5N6 replicated in MDCK and A549 cells, but rH5N6-310PB2 and rH5N6-IG did not replicate in MDCK or A549 cells (Figure 1).

### 3.3. Immunogenicity and Protective Efficacy of Inactivated CD8+ T Cell Epitope-Matched Recombinant H5N6 Strains in Chickens

The immunogenicity and protective efficacy of inactivated oil-emulsion vaccines prepared with rH5N6-310PB2 and rH5N6-IG were compared in SPF chickens (Table 4). The mean HI titers of rH5N6- and rH5N6-IG-vaccinated chickens at 3 wpv were 118.5 and 64.0, respectively, which were unexpectedly low considering their virus titers in the ECEs. Both vaccines completely protected chickens from lethal challenge of the virulent H5N6 strain, and the mean HI titers of the rH5N6- and rH5N6-IG-vaccinated chickens at 1 wpc were 118.5 and 80.6, respectively. The mean HI titer of the rH5N6-vaccinated chickens was not changed after challenge, but that of the rH5N6-IG-vaccinated chickens increased. The increased HI titer may reflect incomplete humoral immunity induction by the rH5N6-IG vaccine. Virus shedding in the oro-pharynx and cloaca was detected in the rH5N6-310PB2-vaccinated (3/9 and 2/9, respectively) but not the rH5N6-IG-vaccinated chickens 7 dpc.

### 3.4. Generation and Replication Efficiency of H103Y-Bearing Recombinant H5N6 Strains

To improve the antigen productivity of H5N6 vaccine strains, we generated H103Y-bearing recombinant strains and verified their replication efficiency in ECEs and mammalian cell lines. The virus titers of rH5N6-H103Y, rH5N6-H103Y-310PB2 and rH5N6-H103Y-IG were 9.03 ± 0.31, 9.58 ± 0.14 and 8.92 ± 0.38 EID_50_/mL, and they were not significantly different from each other (*p* > 0.05, Table 3). The replication efficiency of H103Y-bearing recombinant H5N6 strains was not significantly different from that of corresponding parent strains. In contrast to rH5N6-H103Y-310PB2 and rH5N6-H103Y-IG, only rH5N6-H103Y and rPR8 replicated in both MDCK and A549 cells (Figure 1). The virus titer of rH5N6-H103Y was significantly less than that of rH5N6 24 hpi in MDCK cells and significantly less at 24, 48 and 72 hpi in A549 cells than those of rH5N6 and PR8 (*p* < 0.05). Unexpectedly, the H103Y mutation in the clade 2.3.4.4c H5N6 strain decreased the viral replication efficiency of mammalian cells.

### 3.5. Effect of H103Y Mutation on Immunogenicity of Recombinant H5N6 Strains in Chickens and Ducks

The immunogenicity of inactivated rH5N6-310PB2 and rH5N6-H103Y-310PB2 oil-emulsion vaccines was compared in 3-w-o chickens and 2-w-o ducks (Table 5). In the chickens, the mean HI titers of rH5N6-310PB2 were 98.7 and 90.5, and those of rH5N6-H103Y-310PB2 were 172.3 and 152.2 at 3 wpv and 4 wpv, respectively. In the ducks, the mean HI titers of rH5N6-310PB2 were 14.9 and 12.0, and those of rH5N6-H103Y-310PB2 were 20.2 and 18.7 at 3 wpv and 4 wpv, respectively. rH5N6-H103Y-310PB2 showed significantly higher immunogenicity due to more antigen amount than rH5N6-310PB2 in both the chickens and ducks, but the antibody titers of the ducks vaccinated with the same inactivated vaccine were much lower than those of the chickens (*p* < 0.05).

### 3.6. Effect of the H103Y Mutation on Antigen Productivity in ECEs

To shed light on the reason for higher immunogenicity of H103Y-bearing recombinant virus without significant difference in virus titers, we measured the amounts of total viral protein (Table 6). The virus titers (EID_50_/mL) for rH5N6-310PB2 and rH5N6-H103Y-310PB2 were 9.92 ± 0.4 and 9.4 ± 0.14, but their HA titers were 64.0 ± 0.0 and 107.63 ± 0.9, respectively. Total viral protein for these strains was 1325.4 μg/mL and 2008.75 μg/mL, respectively. Additionally, we compared the intensity of the protein bands separated by SDS-PAGE and verified the greater amount of viral protein for rH5N6-H103Y-310PB2 than for rH5N6-310PB2 (Figure 2). The amount of viral protein was coincident with HA titer rather than EID_50_/mL.

### 3.7. Effects of the H103Y Mutation on Heat and pH Stabilities of Recombinant H5Nx Strains

The effects of the H103Y mutation on heat stability of recombinant H5Nx strains rH5N6-310PB2, rH5N6-H103Y-310PB2, rH5N1-310PB2 and rH5N1-H103Y-310PB2 were examined (Figure 3). H103Y increased the heat stability of rH5N6-H103Y-310PB2 and rH5N1-H103Y-310PB2 to preserve hemagglutination activity even after heat treatment at 55 °C for 2 and 6 h, respectively. Although rH5N1-310PB2 showed lower heat stability than rH5N1-H103Y-310PB2, it showed much higher heat stability than rH5N6-310PB2.

The effects of H103Y on the pH stability of the recombinant H5Nx strains rH5N6, rH5N6-H103Y, rH5N6-310PB2, rH5N6-H103Y-310PB2, rH5N1-310PB2, rH5N1-H103Y-310PB2, rH5N8 and rH5N8-H103Y were tested (Figure 4). H103Y decreased the pH stability of rH5N6-H103Y, rH5N6-H103Y-310PB2 and rH5N1-H103Y-310PB2 to cause the loss of infectivity at pH 5.0, 5.0 and 4.8, respectively, but had no effects on the pH stability of rH5N8 or rH5N8-H103Y. Differences in pH stability among the H5Nx strains bearing H103Y were apparent, and the clade 2.3.2.1c H5N1 strain was highly stable and maintained viral infectivity at pH 5.0, compared to the clade 2.3.4.4c H5N6 (pH 5.2) and clade 2.3.4.4a H5N8 (pH 5.4) strains.

## 4. Discussion

To date, clade 2.3.4.4c H5N6 HPAIVs have caused 22 human cases and more frequent human and mammalian infections than other clade 2.3.4.4 H5Nx HPAIVs, emphasizing the importance of the preparation of safer vaccine strains and greater vaccine preparedness [29]. Conventional PR8-derived recombinant vaccine strains contain 6 internal genes encoding 10 internal proteins (PB2, PB1, PB1-F2, PA, PA-X, NP, M1, M2, NS1 and NEP) with mammalian pathogenicity-related mutations [30,31]. Among these, the crucial mutations, such as E627K, Q591R/K and D701N in PB2 and the PKR (Protein kinase R)-binding site and the PDZ domain in NS1, are very influential for the acquisition of mammalian pathogenicity of AIVs [12,32,33]. To reduce the unintended in vitro mammalian pathogenicity acquisition by a conventional reverse genetics system, we successfully applied 01310 PB2 and 0028 NS genes to generate safer PR8-derived recombinant vaccine strains [13]. As expected, 01310 PB2 in rH5N6-310PB2 preserved the viral replication efficiency in ECEs and eliminated the viral replication capacity in MDCK and A549 cells in this study (Table 3 and Figure 1). The 0028 NS mutation improved viral yield in the ECEs by reducing the embryonic pathogenicity of a recombinant H9N2 strain [13]. None of the recombinant H5N6 strains bearing single (310PB2) or multiple (310 PB2, NP, M, and NS) avian internal genes caused embryonic death within 3 days, and we were not able to verify the individual effect of 0028 NS in this study.

In contrast to formaldehyde, BPL-inactivated oil-emulsion vaccine induced cellular immunity via specific CD8+ T cells activated by epitopes presented by class I MHC molecules [13,14]. As BEI inactivates viral RNA rather than proteins, we have used BEI for the preparation of inactivated vaccines to reduce the deformation of antigenic structures. Although we did not demonstrate cytoplasmic penetration of the RNPs and M1 proteins of BEI-inactivated H5N6 vaccine strains, BEI-inactivation is unlikely to hinder viral endocytosis or endosomal fusion required for the processing and presentation of CD8+ T cell epitopes. Although inactivated oil-emulsion vaccines rH5N6-310PB2 and rH5N6-IG prevented mortality after lethal challenge with the homologous subtype strain, the efficacy of the vaccine in protecting against virus shedding may be incomplete. However, earlier cessation of virus shedding in rH5N6-IG-vaccinated chickens with an even lower HI titer can be explained by the induction of cellular immunity to compensate for incomplete humoral immunity. Considering the greater number of matched CD8+ T cell epitopes for NP between rH5N6-IG and the challenge virus, efforts to select optimal NP genes in terms of matching of CD8+ T cell epitopes may be important to improve vaccine efficacy. As we were not able to compare the cellular immunity induced by the rH5N6-310PB2 and rH5N6-IG vaccinations, we will confirm the roles of the predicted CD8+ T cell epitopes for NP and the induction of specific CD8+ T cells in further studies. Due to the complicated interactions of NP with PB1 and PB2 and of M1 with viral ribonucleoprotein and viral transmembrane proteins (HA, NA and M2), optimal NP and M genes need to be selected for better viral fitness of vaccine strains [34,35,36,37]. For this reason, we have not been able to optimize M genes to match CD8+ T cell epitopes in M1 or a B cell epitope in M2e at this time.

The effects of H103Y on viral replication efficiency in ECEs and heat stability of vaccine strains were reverified in this study [13]. In contrast to clade 2.3.4.4a H5N8 strains, the previous experimental results that PR8-derived recombinant clade 2.3.4.4c H5N6 strains had sufficiently high virus titers without any genomic modification did not encourage efforts to increase amount of clade2.3.4.4c antigen any further [38]. However, the addition of H103Y to form rH5N6-H103Y-310PB2 significantly increased the amount of antigen in this study (Figure 2). Therefore, checking antigen amount versus infectious unit may be important. When the H103Y mutation was adopted in clade 2.3.4.4c H5 protein, it interacted with other residues (arginine at residue 75 and asparagine at 79 of HA2) of other HA monomers by polar contact (Appendix A). This intermolecular interaction between HA trimers increased structural stability and maintained conformational structure even after heat treatment. The heat stability of vaccine strains is important for vaccine production and storage, and H103Y may be useful for extending the shelf-lives of vaccine products.

The unexpected effects of H103Y on decreased (clade 2.3.4.4c H5N6 and clade 2.3.2.1c H5N1 strains) or unchanged (clade 2.3.4.4a H5N8 strain) low pH stability was contradictory to previous reports (Figure 4). In previous reports, the H103Y in H5N1 strains (A/Indonesia/5/2005(H5N1)) (Indo5) showed increased low-pH stability, resulting in decreasing activation pH of HA and increased human pathogenicity via the facilitation of survival in the acidic upper respiratory tract and horizontal transmission [21,27,39]. As the pKa of the side chain is 6.0, histidine plays a role as a sensor of HA at an activation pH of approximately 5.0 to 6.0 [40,41]. Therefore, the number and locations of histidine residues may affect activation pH and stability of HA. The rH5N6 and rH5N8 (and rH5N1) strains used in this study have 2 (1) more histidines at 240 and 273 (H3 numbering) than for Indo5, showing a different effect for H103Y. H240 on the interfaces of the globular heads of HA trimers probably participates in intermolecular interaction with S217 of neighboring HA monomers via H-bonding (Appendix A). However, different effects of H103Y mutations in H5N6 and H5N8 strains on low pH stability and replication efficiency in MDCK and A549 cells suggest the possible involvement of different NA subtypes. The different subtypes of NA differed in activity with mammalian receptor (α-2,6-sialogalactose), and 11aa- and 23aa-deleted stalks of N6 and N1 may act differently from N8 with the stalk intact [42]. In previous reports, different NA subtypes have been found to affect the activation pH of HA differently [39]. In any case, H103Y in the genetic background of the clade2.3.4.4c H5N6 strain reduced replication efficiency in mammalian cells, reflecting decreased mammalian pathogenicity.

## 5. Conclusions

BEI-inactivated vaccine possessing similar CD8+ T cell epitopes of NP to those of the challenge virus halted virus shedding earlier, and the H103Y mutation in H5Nx viruses consistently resulted in increased heat stability but different pH stability under differing genetic backgrounds of HA and NA genes. Thus, efforts to improve and optimize the conventional PR8-based reverse genetics system need to be intensified to develop better avian influenza vaccines in the near future.

## Figures and Tables

**Figure 1 vaccines-08-00781-f001:**
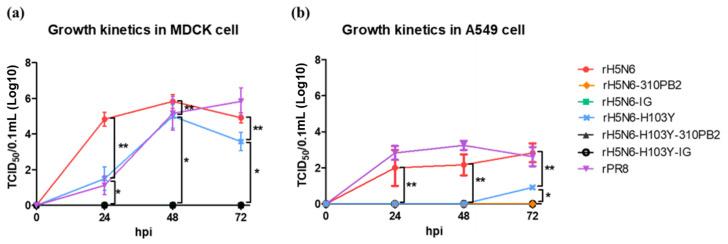
Growth kinetics in Madin-Darby canine kidney (MDCK) and A549 cells of recombinant H5N6 viruses. 5 × 10^5^ EID_50_/0.5 mL of each recombinant viruses were infected into (**a**) MDCK and (**b**) A549 monolayer in 12 well plate. The infected virus was aspirated after 1 h and 1 mL of fresh medium was added to measure the viral titer in the supernatant obtained every 24 h. TCID_50_/0.1 mL of virus in supernatant of each time-point were measured by inoculation of serial 10-fold diluents into MDCK cells in 96 well plate. Each result was the average of three independent experiments. Statistical differences of rH5N6-H103Y to the other viruses were marked with * (*p* < 0.05) and that of rH5N6 were with ** (*p* < 0.05).

**Figure 2 vaccines-08-00781-f002:**
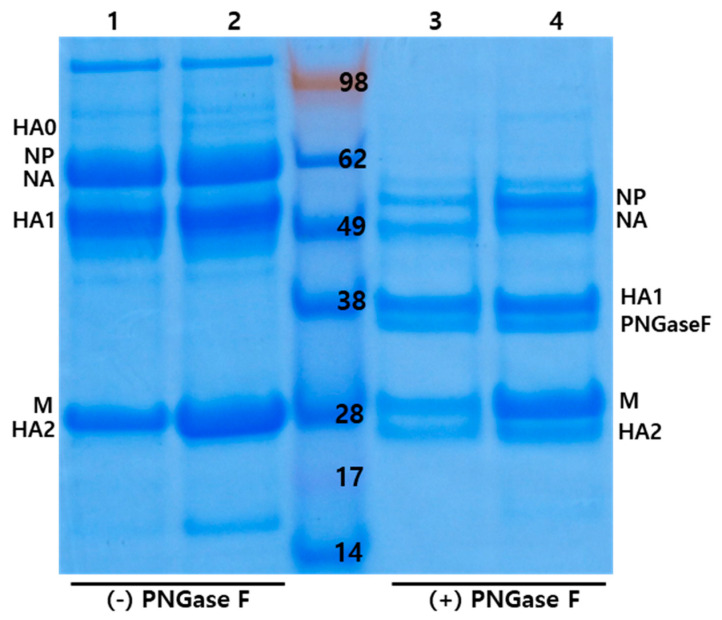
SDS-PAGE of purified total viral protein of recombinant H5N6 viruses. Purified total viral protein of rH5N6-310PB2 (lane 1, 3) and rH5N6-H103Y-310PB2 (lane 2, 4) acquired by ultracentrifugation were separated by SDS-PAGE. Deglycosylation of viral proteins was conducted using PNGase F and deglycosylated protein were also separated by SDS-PAGE (lane 3, 4). Lane 1; total viral protein of rH5N6-310PB2, lane 2; total viral protein of rH5N6-H103Y-310PB2, lane 3; PNGase F treated total viral protein of rH5N6-310PB2 and lane 4; PNGase F treated total viral protein of rH5N6- H103Y-310PB2.

**Figure 3 vaccines-08-00781-f003:**
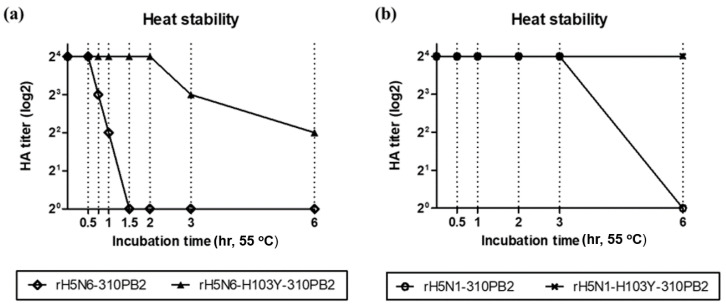
Stability of HA protein after heat treatment. 16 (2^4^) HAT aliquots of each viruses were incubated at 55 °C during different incubation times. After incubation, HAT of heat-treated viruses were re-measured using 1 % chicken RBC. (**a**) rH5N6-310PB2 and rH5N6-H103Y-310PB2 virus (**b**) rH5N1-310PB2 and rH5N1-H103Y-310PB2 virus showed different resistance in high temperature. rH5N1 viruses (clade 2.3.2.1c HA protein) showed relatively higher stability at 55°C than rH5N6 viruses (clade 2.3.4.4c HA protein). The H103Y mutants in both strains acquired heat stability, sustaining HA activity after heat treatment.

**Figure 4 vaccines-08-00781-f004:**
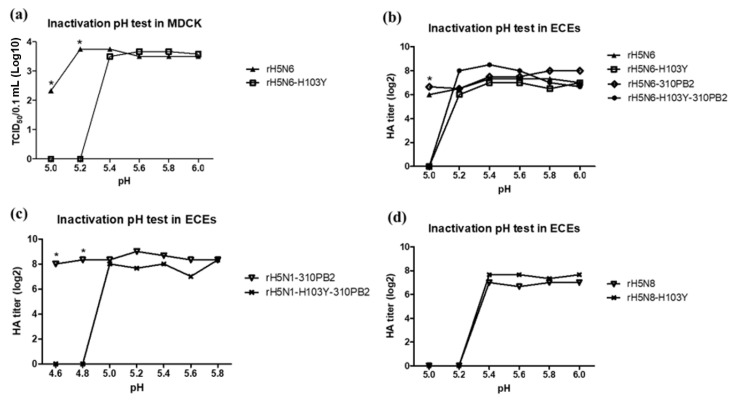
Inactivation at low pH of recombinant H5N1, H5N6 and H5N8 viruses. 5 × 10^5^ EID_50_/0.5 mL and 10^4^ EID_50_/0.1 mL of virus was treated with pH buffer for 1 h and inoculated into (**a**) MDCK cells and (**b**–**d**) 10 day-old SPF ECEs (**b**). After 72 h incubation, cell supernatant and allantoic fluids were obtained and tissue culture infective dose (TCID_50_)/0.1 mL in supernatant and HA titer of allantoic fluid were measured. Data was presented as average of three independent replicates. The statistical significance was indicated by asterisks (*p* < 0.05).

**Table 1 vaccines-08-00781-t001:** Mutagenesis primer sets used in H103Y mutation of recombinant H5N6 and H5N1 viruses.

Primer Name	Sequence (5′–3′)
H5N6-H103Y-F	*TATGAAGAACTGAAATACCTATTGAGCAGAA*
H5N6-H103Y-R	*TTCTGCTCAATAGGTATTTCAGTTCTTCATA*
H5N1-H103Y-F	*TATGAAGAATTGAAATACCTATTGAGCAGGA*
H5N1-H103Y-R	*TCCTGCTCAATAGGTATTTCAATTCTTCATA*

**Table 2 vaccines-08-00781-t002:** Comparison of expected T cell epitopes sequences in NP and M genes of challenge virus with other highly pathogenic avian influenza A viruses (HPAIVs) (2018–2020 isolates) and genomes used in this study.

	NP							M			
	10–17	26–34	76–83	127–134	249–258	347–354	368–375	134–142	162–169	200–207	229–237
	YEQMETGG ^b,c^	RASVGRMVG	NRYLEEHP	EDATAGLT	GNAEIEDLIF	IRGTRVVP	NENMETMD	RMGTVTTEV	HRQMATIT	AEAMEVAN	LRDNLLENL
H5N1 (*n* = 21)	........ ^a,^^d^	......... .......S.	........ .K......	........	..........	........ ...A....	........ ...T....	.........	........ ...I..T.	.....I..	.........
H5N8 (*n* = 26)	........	.........	.K......	........	..........	........	........	......... K........	...I..T.	.......S	.........
H5N6 (*n* = 37)	........	.........	........ .K......	........	..........	........ ...A.... ......F.	........	.........	........ ...I..T.	........ .......S	......... ...N.....
SNU50-5 (H5N1)	........	.........	.K......	........	..........	........	........	.........	....V.T.	.......S	.K.D.....
01310 (H9N2)	........	.........	........	........	..........	...A....	...T....	.........	....V.T.	.......S	.K.D.....
0028 (H9N2)	........	.........	........	........	..........	........	...T....	.........	....V...	.....I.S	.K.D.....
PR8	......D.	.....K.I.	.K......	D.......	....F...T.	.K..K.L.	.......E	...A.....	....V.T.	.......S	.KND.....

^a^ only epitopes with sequence differences between viruses were shown. ^b^ peptide sequence of challenge virus (A/Mandarin_duck/Korea/K16-187-3/2016). ^c^ anchor residues recognized by chicken MHC is underlined. ^d^ same amino acid with the peptide sequence was denoted with dot.

**Table 3 vaccines-08-00781-t003:** Genomic composition and replication efficiency in embryonated chicken eggs (ECEs) of generated recombinant H5N6 strains.

Recombinant Virus	HA	NA	PB2	PB1	PA	NP	M	NS	EID_50_/mL ^a^
rH5N6	H5	N6	PR8	PR8	PR8	PR8	PR8	PR8	9.08 ± 0.14
rH5N6-310PB2	H5	N6	01310	PR8	PR8	PR8	PR8	PR8	9.33 ± 0.29
rH5N6-IG	H5	N6	01310	PR8	PR8	SNU50-5	01310	0028	9.25 ± 0.25
rH5N6-H103Y	H5-H103Y	N6	PR8	PR8	PR8	PR8	PR8	PR8	9.03 ± 0.31
rH5N6-H103Y-310PB2	H5-H103Y	N6	01310	PR8	PR8	PR8	PR8	PR8	9.58 ± 0.14
rH5N6-H103Y-IG	H5-H103Y	N6	01310	PR8	PR8	SNU50-5	01310	0028	8.92 ± 0.38

^a^ Embryo infectious dose (EID_50_)/_mL_ of harvested virus after 100 EID_50_ of each recombinant virus inoculation into 10 day-old specific-pathogen-free (SPF) ECEs. Average ± standard deviation (SD) of three independent replicates.

**Table 4 vaccines-08-00781-t004:** Serum antibody titers, survival rates and viral shedding rates of chicken groups vaccinated with inactivated recombinant H5N6 viruses.

Inactivated Vaccine Strain	GMT of HI Titer ^a^			Survival Rate	Viral Shedding Rate ^c^							
	0 wpv	3 wpv ^b^	1 wpc		Oro-pharyngeal Swab				Cloacal Swab			
					1 dpc	3 dpc	5 dpc	7 dpc	1 dpc	3 dpc	5 dpc	7 dpc
rH5N6-310PB2	<2 ^d^	118.5 (78.14–179.7)	118.5 (72.3–194.3)	9/9 (100%)	2/9	7/9	4/9	3/9	2/9	6/9	4/9	2/9
rH5N6-IG	<2	64.0 (43.9–93.3)	80.6 (55.3–117.5)	9/9 (100%)	4/9	4/9	4/9	0/9	1/9	5/9	4/9	0/9
Mock	<2	<2	<2	0/9 (0%)	9/9	9/9	nt ^e^	nt	7/9	9/9	nt	nt

^a^ Geometric mean hemagglutination inhibition (HI) titer with 95 % confident interval. ^b^ wpv; week-post-vaccination, wpc; week-post challenge. ^c^ Real-time RT-PCR results of RNA extracted from oro-pharyngeal and cloacal swab sample. ^d^ Undetectable level of serum antibody. ^e^ nt; not tested.

**Table 5 vaccines-08-00781-t005:** Effect of H103Y mutation on immunogenicity of hemagglutinin (HA) in chickens and ducks.

Species	Vaccination Age	Inactivated Vaccine Strain	GMT of HI Titer ^a^		
			0 wpv ^b^	3 wpv	4 wpv
Chicken	3 week-old	rH5N6 -310PB2	<2 ^c^	98.70 ^†^ (64.13–151.9)	90.51 ^†^ (42.38–193.3)
		rH5N6-H103Y-310PB2	<2	172.3 ^†^ (88.28–356.4)	152.2 ^†^ (68.06–340.4)
		Control	<2	<2	< 2
Duck	2 week-old	rH5N6 -310PB2	<2	14.86 (6.95–22.77)	12.00 (7.22–16.78)
		rH5N6-H103Y-310PB2	<2	20.16 (9.49–42.83)	18.66 * (9.84–35.41)
		Control	<2	<2	<2

^a^ Geometric mean HI titer with 95 % confident interval. ^b^ wpv; week-post-vaccination. ^c^ Undetectable level of serum antibody. ^†^ Significantly different with serum HI titer of paired duck vaccination groups (*p* < 0.05). * Significantly different with serum HI titer of rH5N6-310PB2 vaccinated duck (*p* < 0.05).

**Table 6 vaccines-08-00781-t006:** Effect of H103Y mutation of HA on virus titers and antigen amount.

	EID_50_/_mL_ (log10) ^a^	HA Titer ^b^	Amount of Virus Total Protein (μg/mL) ^a^
rH5N6-310PB2	9.92 ± 0.38	64.00 ± 0.00	1325.42
rH5N6-H103Y-310PB2	9.42 ± 0.14	107.63 ± 0.89	2008.75

^a^ EID_50_ and total protein amount of pooled allantoic fluid after 100 EID_50_/0.1 mL of each viruses inoculation into eight 10-day-old SPF ECEs. ^b^ Average HA titer of each harvested allantoic fluids (± SD).

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
