# Peer review of "Improvement of PR8-Derived Recombinant Clade 2.3.4.4c H5N6 Vaccine Strains by Optimization of Internal Genes and H103Y Mutation of Hemagglutinin"

_vaccines, 2020, doi:10.3390/vaccines8040781_

Round 1

Reviewer 1 Report

The authors aim to improve current PR8-based vaccines by insertion of genes from AIVs into the PR8-derived H5N6 vaccine strain to improve antigen productivity, immunogenicity, vaccine protection and antigen stability while retaining low mammalian pathogenicity. They assess multiple modified recombinant vaccine strains including those with a modified PB2 gene, NP gene, M gene or H103Y HA mutation. Genes are chosen from AIVs based on the presence of potential chicken CD8+ T cell epitopes. Some recombinant strains have multiple gene changes. They assess growth kinetics in ECEs as an assessment of antigen productivity and viral fitness as well as in canine and human cell lines as an assessment of mammalian pathogenic potential. They also assess a number of the modified vaccines strains in chicken and duck challenge experiments.

Overall, this is a worthwhile, well written and nicely presented collection of data. However, I have the following concerns:

Major:

  • The addition of the H103Y mutation resulted in greater overall virus protein antigen production (albeit not infectious virus) as indicated in Table 6 and Figure 2. Thus, higher immune responses in Table 5 do not necessarily equate to overall immunogenicity of the vaccinating strain and should not be represented as such. They may simply be due to a higher overall dose of vaccine protein antigen. It is important that the authors make this clear and do not misconstrue this higher dose of overall vaccinating proteins with greater immunogenicity of the specific genes in the modified vaccine strain. If the authors wish to ascertain differences in genuine immunogenicity of each strain, they need to vaccinate with equivalent protein amounts of each vaccine strain and then assess outcomes. 
  • Based on the findings mentioned in the previous comment. Were overall protein antigen levels administered in the challenge study in Table 4 also ascertained? Might the “unexpectedly low” HI titres of rH5N6-IG be simply due to reduced levels of overall protein compared to the 310PB2 strain?
  • If initial live virus to dead virus ratio of the ultimately inactivated and inoculated vaccine virus is important in vaccine potency (as possibly suggested by the data discussed in the previous comment) this is also worth evaluating more closely. 
  • Conclusions/claims about increased immunogenicity in the manuscript need to be tempered and described more clearly based on these observations. Discussion of vaccine potency based on initial infectious units versus total protein antigen would be more accurate.

Minor:

  • There need to be error bars on the graphs in Figure 1.
  • The use of * and ** in Figure 1 is confusing. Points that appear more different have been assigned lower significance and the legend merely defines them both as <0.05. Is ** actually 0.01? Please make clearer.
  • In Table 4 move the b superscript onto “3 wpv”. Under the e superscript write out: “nt; not tested”

Author Response

Thank you for your comments

[Reviewer 1]

Major:

  • The addition of the H103Y mutation resulted in greater overall virus protein antigen production (albeit not infectious virus) as indicated in Table 6 and Figure 2. Thus, higher immune responses in Table 5 do not necessarily equate to overall immunogenicity of the vaccinating strain and should not be represented as such. They may simply be due to a higher overall dose of vaccine protein antigen. It is important that the authors make this clear and do not misconstrue this higher dose of overall vaccinating proteins with greater immunogenicity of the specific genes in the modified vaccine strain. If the authors wish to ascertain differences in genuine immunogenicity of each strain, they need to vaccinate with equivalent protein amounts of each vaccine strain and then assess outcomes. 

> We agree with reviewer’s comment and lower immunogenicity may be only due to less amount of vaccine protein antigen. We revised the manuscript as recommended by omitting misconstrueing ‘immunogenicity’.

- Abstract, we removed in line 29: ‘and immunogenicity’; in line 35: ‘, immunogenic’.

- Introduction, the line 85: we replaced ‘immunogenicity’ with ‘antigen productivity’; the lines 81: we omitted ‘, immunogenicity’.

- Result 3.4 the line 2, ‘immunogenicity’ to ‘antigen productivity’; 3.5, the line 19: we added ‘due to more antigen amount’

  • Based on the findings mentioned in the previous comment. Were overall protein antigen levels administered in the challenge study in Table 4 also ascertained?

    > No, we did not.

  • Might the “unexpectedly low” HI titres of rH5N6-IG be simply due to reduced levels of overall protein compared to the 310PB2 strain?

> To our experimental experience the HI titers of serum samples from both rH5N6- and rH5N6-IG-vaccinated chcikens were low in comparison with their virus titers. We corrected ‘was’ to ‘were’ not to mislead our intention of description in line 32 of Result 3.4. The relatively lower HI titer of rH5N6-IG may be due to less antigen amount as recommended.

  • If initial live virus to dead virus ratio of the ultimately inactivated and inoculated vaccine virus is important in vaccine potency (as possibly suggested by the data discussed in the previous comment) this is also worth evaluating more closely. 

> Yes, we completely agree with reviewer’s comment and we revised the manuscript as in line 19 of Result 3.5.

  • Conclusions/claims about increased immunogenicity in the manuscript need to be tempered and described more clearly based on these observations. Discussion of vaccine potency based on initial infectious units versus total protein antigen would be more accurate.

> Yes, we completely agree with reviewer’s comment and we revised the manuscript as in above and we added ‘Therefore, checking antigen amount versus infectious unit may be important.’ In the line 119-120 of Discussion section.

Minor:

  • There need to be error bars on the graphs in Figure 1.

    > We added error bars (for standard deviation of results) in Figure 1.

  • The use of * and ** in Figure 1 is confusing. Points that appear more different have been assigned lower significance and the legend merely defines them both as <0.05. Is ** actually 0.01? Please make clearer.

> * represented statistical differences of rH5N6-H103Y to other viruses and ** showed statistical differences of rH5N6 with others. We revised the legend of Figure 1 into “Statistical differences of rH5N6-H103Y to the other viruses were marked with * (p <0.05) and that of rH5N6 were with ** (p <0.05).”

  • In Table 4 move the b superscript onto “3 wpv”. Under the e superscript write out: “nt; not tested”

    > The b was moved onto “3 wpv” and e Not tested was modified into e nt; not tested.

Reviewer 2 Report

The authors tried to improve conventional PR8-derived recombinant 2.3.4.4c H5N6 vaccine strain in terms of antigen productivity, reduced mammalian pathogenicity, immunogenicity, protection efficacy, and antigenic stability. This is a well written, interesting, and useful contribution, which I think is entirely suitable for publication in Vaccines. However I do see the need for some clarifications and smaller corrections.

Major points;

P 3, Line 122: Please add more description for the full genome sequences. The readers can not understand the method you used.

P 4, Line 142: Please indicate the supplier and Cat. No. for “binary ethylenimine and sodium thiosulfate“ so that people can reproduce your experiment.

P 4, Line 156: Your HI test method does not much with the description in WHO Manual on Animal Influenza Diagnosis and Surveillance. Please confirm.

P 6, Table 2: Footnote. Where are a, b and e?

Minor points;

P 3, Line 120: Delete “t”.

P 14, Line 127: A/Indonesia/5/05 should be A/Indonesia/5/2005

P 14, Line 139: α is missing.

Author Response

Thank you for your comments.

[Reviwer 2]

Major points;

P 3, Line 122: Please add more description for the full genome sequences. The readers can not understand the method you used.

> We added detailed description of procedure conducted for full genome sequencing in line 124 – 127. “Briefly, RNA was extracted from harvested allantoic fluid using Viral Gene-spin™ Viral DNA/RNA Extraction Kit (iNtRON Biotechnology) and cDNA was synthesized by TOPscript™ cDNA Synthesis kit (Enzynomics, Daejeon, Korea). Full genomes were amplified using universal primer sets previously described [23].”

P 4, Line 142: Please indicate the supplier and Cat. No. for “binary ethylenimine and sodium thiosulfate“ so that people can reproduce your experiment.

> BEI was made by mixing 2-bromoethlyemine hydroxide and NaOH purchased from Sigma-Aldrich, and description was added in line 147 – 149, “.1 M binary ethylenimine (BEI) prepared by mixing 0.041g of 2-bromoethylamine hydrobromide (Sigma-Aldrich) with 2ml of 0.175N sodium hydroxide (Sigma-Aldrich)”.

P 4, Line 156: Your HI test method does not much with the description in WHO Manual on Animal Influenza Diagnosis and Surveillance. Please confirm.

> We basically followed the manual but modified some details that was explained by “HI assays were performed based on the WHO Manual on Animal Influenza Diagnosis and Surveillance with modification.” In line 166-167.

P 6, Table 2: Footnote. Where are a, b and e?

> a was in the first column of the H5N1 line, and b was added in the first T cell epitope(YEQMETGG) of NP. e was in supplementary data, but accidently added and deleted.

Minor points;

P 3, Line 120: Delete “t”.

> The typo might be edited by journals.

P 14, Line 127: A/Indonesia/5/05 should be A/Indonesia/5/2005

> The strain name in line 127 was revised into “A/Indonesia/5/2005”

P 14, Line 139: α is missing.

> That typo was fixed into “α -2,6-sialogalactose”.

Round 2

Reviewer 1 Report

Thank-you for addressing my concerns. I have only one further minor correction:

Discussion Line 118:

Not entirely sure what is being said in the new highlighted version. I think the authors mean: "did not encourage efforts to increase antigen amount ANY further". Please revise to be more clear.

Author Response

  • Discussion Line 118:

Not entirely sure what is being said in the new highlighted version. I think the authors mean: "did not encourage efforts to increase antigen amount ANY further". Please revise to be more clear.

  • We agreed with your opinion and added more detailed explanation, “In contrast to clade 2.3.4.4a H5N8 strains, the previous experimental results that PR8-derived recombinant clade 2.3.4.4c H5N6 strains had sufficiently high virus titers without any genomic modification did not encourage efforts to increase amount of clade2.3.4.4c antigen any further [38].”